# *Morganella* Phage Mecenats66 Utilizes an Evolutionarily Distinct Subtype of Headful Genome Packaging with a Preferred Packaging Initiation Site

**DOI:** 10.3390/microorganisms10091799

**Published:** 2022-09-07

**Authors:** Nikita Zrelovs, Juris Jansons, Andris Dislers, Andris Kazaks

**Affiliations:** Latvian Biomedical Research and Study Center, Ratsupites 1 k-1, LV-1067 Riga, Latvia

**Keywords:** *Morganella*, bacteriophage, complete genome, headful packaging, packaging initiation (*pac*) site, genome termini, terminase

## Abstract

Both recognized species from the genus *Morganella* (*M. morganii* and *M. psychrotolerans*) are Gram-negative facultative anaerobic rod-shaped bacteria that have been documented as sometimes being implicated in human disease. Complete genomes of seven *Morganella*-infecting phages are publicly available today. Here, we report on the genomic characterization of an insect associated *Morganella* sp. phage, which we named Mecenats66, isolated from dead worker honeybees. Phage Mecenats66 was propagated, purified, and subjected to whole-genome sequencing with subsequent complete genome annotation. After the genome de novo assembly, it was noted that Mecenats66 might employ a headful packaging with a preferred packaging initiation site, although its terminase amino acid sequence did not fall within any of the currently recognized headful packaging subtype employing phage (that had their packaging strategy experimentally verified) with clusters on a terminase sequence phylogenetic tree. The in silico predicted packaging strategy was verified experimentally, validating the packaging initiation site and suggesting that Mecenats66 represents an evolutionarily distinct headful genome packaging with a preferred packaging initiation site strategy subtype. These findings can possibly be attributed to several of the phages already found within the public biological sequence repositories and could aid newly isolated phage packaging strategy predictions in the future.

## 1. Introduction

Given their ubiquity and abundance, multiple bacteriophages infecting any bacterial host are expected to exist in nature, however, currently, the known cultured phage diversity is still largely restricted to a relatively limited number of distinct hosts. As the potential of phages to combat pathogenic bacteria was clear even prior to the discovery of the first antibiotics, understandably, the currently available cultured phage diversity mostly comprises viruses infecting bacterial pathogens of importance to humans, studies of which have gained renewed interest in light of the emergence of antimicrobial resistance. Nowadays, phages are viewed either as a promising alternative or as a complement to antibiotics to treat bacterial diseases, however, there are still bacterial genera including opportunistic pathogens of arguably growing concern, for which either a very limited number of phages have been acquired in culture, sequenced, and described in detail (e.g., *Morganella* [1], *Tsukamurella* [2], *Brevundimonas* [3], *Kluyvera* [4]), or none at all yet (e.g., *Alloscardovia*, *Capnocytophaga*, and a number of intracellular bacterial pathogens such as *Rickettsia*, among others [5]). However, metagenome-assembled phages predicted to infect members of the bacterial genera that do not yet have known cultured and described phages that are able to infect them as well as/or prophage/prophage remnant regions in the genomes of at least some of these bacteria have been previously identified [6,7,8].

The bacterial genus *Morganella* of the *Enterobacteriaceae* family was for a long time thought to comprise a single species *M. morganii*, which is represented by two subspecies (*M. m. morganii* and *M. m. sibonii*), representatives of which have previously been isolated from a variety of both environmental and higher-organism-associated sources, according to the metadata of the publicly available *Morganella* spp. derived sequences found in GenBank. Although *M. morganii* is considered as a commensal intestinal tract bacterium of humans and other mammals, with the infections caused by this opportunistic pathogen previously being infrequent, the wide disease spectrum associated with *M. morganii* as well as the increasing drug resistance of *M. morganii* isolates make it a pathogen that should not be neglected [1]. Additionally, the bacterial genus *Morganella* expanded with a novel species in 2006, *Morganella psychrotolerans* [9], a type strain of which was isolated from cold-smoked tuna involved in an outbreak of histamine poisoning in Denmark in 2004 [10]. Thus, isolation, propagation, and subsequent characterization of *Morganella* phages, which could eventually lead to the availability of *Morganella*-infecting phage stocks that can be rapidly amplified in the case of need, might be beneficial in the future.

To date, at least seven phages of *Morganella* sp. have had their complete genomes elucidated and deposited in public biological sequence repositories (Table 1). Thus, the currently sequenced *Morganella* phages include the phages MmP1 (EU652770; [11]), MP1 and MP2 (KX078569 and KX078568, respectively; [12]), Lilipawes and Rgz1 (OK499982 and OK499989, respectively; [13]) as well as IME1369_01 (KY653118) and IME1369_002 (KY653119). Five out of these seven *Morganella* phages sequenced thus far have been isolated from sewage, arguably the most common phage isolation source, whereas two of them (phages IMEI1369_1 IMEI1369_2) were found to be prophages of the *M. morganii* strain IMEI1369 without an article describing them. The literature, however, lists at least one more phage that is capable of infecting bacteria from the genus *Morganella* that has not yet been characterized genomically—the phage FSP1 isolated from river water [14].

Tailed dsDNA-containing bacteriophages were shown to package their genomic DNA into the assembled procapsids using a molecular motor usually consisting of three different proteins: portal protein and terminase small (TerS), and large (TerL) subunits [15,16]. The portal protein forms a channel through which the phage DNA is packaged inside the procapsid during the virion assembly, and is later released outside a mature capsid when an assembled intact virion recognizes a suitable host and initiates the infection process [17]. The terminase binds to the portal ring, recognizes the phage DNA, and translocates the phage genome inside the prohead utilizing ATP hydrolysis before any further viral particle assembly can proceed [16,18].

For most tailed dsDNA phages, replication of the genetic material is believed to result in the formation of the DNA concatemers, representing multiple copies of the phage genome linked one after the other [19], which serve as a genome packaging substrate (however, other packaging substrates were also documented, e.g., [20,21]). These phages have evolved their own (rather than host), multiple, DNA recognition mechanisms as well as genome packaging initiation and termination strategies that result in a variation among the physical genome molecule termini within the capsids of different phages. These include precise DNA termini with defined cohesive (*cos*) 5′ [22] or 3′ [23] overhangs, and fixed direct terminal repeats of varying lengths (either “short” [24] of up to several hundred base pairs or “long” [25] of up to several thousand base pairs).

Packaging strategies that do not imply the same precise genome termini in all of the progeny virions are also frequent. The so-called “headful packaging”-employing phage [26] progeny has more than a single genome length packaged inside the procapsid, which makes the genomes of the progeny virions terminally redundant and circularly permuted when compared to each other [27]. Headful packaging-employing phages fill the procapsid with genomic DNA until it is full, and depending on a particular packaging strategy, can either recognize a packaging series initiation site (*pac*) on a phage genome concatemer and package more than a single copy of the genome inside the procapsid [28] or initiate the packaging seemingly randomly [29]. It is, however, worth noting that not all of the tailed dsDNA phages employ the aforementioned packaging strategies, and a number of more “exotic” phage genome packaging strategies have been previously identified (e.g., host DNA at the phage genome termini [21], proteins covalently attached to the phage genome termini [30]).

The accumulation of different unrelated phage genomes sequenced throughout the years and elucidation of their genome termini types/packaging strategies have revealed that the packaging strategy a phage employs can be predicted in silico using either NGS read pile-up analyses or TerL amino acid sequence phylogeny reconstructions that include phages with packaging strategies determined previously, however, ideally, both approaches should be undertaken as they complement each other very well [31,32,33,34]. While the read pile-up analyses are more mechanistic and are expected to work in the case of correctly prepared NGS libraries (without violating the assumptions of the methods), the extent of the applicability of the TerL aa sequence-based phylogeny approach for the prediction of packaging strategy for novel phages is currently unclear, although attempts to resolve the complex phylogenetic relationships of phage or phage-like element terminases/TerLs might help to shed light on this matter [35]. Therefore, this approach might not provide unambiguous results for novel phages whose TerL sequences are extremely divergent from their counterparts, encoded by phages whose genome termini types have been elucidated experimentally. While the genome termini determination aspect is largely ignored in the majority of the recently publicly submitted phage genomes, generalized experimental methods that allow for a comparatively fast and easy determination of the phage genome termini type have been established [27].

In this study, the genomic characterization of a novel bacteriophage “Mecenats66” isolated from a dead worker honeybee is described. The host of Mecenats66, isolated from the same insect specimen, was identified as a species belonging to the bacterial genus *Morganella*, putatively representing a novel bacterial species within that genus, based on its near full-length 16S rRNA gene sequence phylogeny. Upon obtaining the complete genome sequence of phage Mecenats66, it was noted that it might employ an evolutionary distinct subtype of the headful packaging strategy, which was further verified experimentally. Additionally, a glimpse into the representative diversity of the TerL/terminase amino acid sequences is given, showing that the dataset comprising some of the most well-studied phages for which genome termini/packaging strategies are known fails to accommodate for the complex evolutionary relationships of TerL/terminase proteins among different packaging strategy employing phages.

## 2. Materials and Methods

### 2.1. Host Bacteria and Phage Isolation and Propagation

Both the phage and its host bacteria were isolated from dead bees collected at a bee farm located in a private field in Limbaži Municipality near Pociems, Latvia, as described previously [36], with a modification that LB agar (g/L: Bacto-Tryptone—10, Bacto yeast Extract—5, NaCl—10, Agar—15) was used for both the isolation of bacteria and as a bottom agar in a standard double-agar overlay (with 0.7% LB as top agar) using the initially retrieved morphologically distinct colonies (that were subcultured twice to obtain indicator cultures for bacteriophage isolation). Plates with 10–100 μL of the crushed insect suspension supernatant were incubated for 1 to 3 days at RT to obtain individual bacterial colonies. The working indicator culture on which phage Mecenats66 was later found was designated as B1-1.

Several individual negative colonies that were found in a double agar overlay on the lawn of an indicator culture B1-1 using 50 to 100 μL of the crushed bee suspension filtrate were purified by plaque subculturing twice. One of the plaques was further propagated through the confluent lysis double-agar overlay plates several times. Briefly, after incubation overnight at RT, the top agar material from several plates was collected and homogenized by vortexing with the addition of a minimal amount of LB, clarified by centrifugation (12,000× *g*, 20 min) and filtered through a 0.45 μm pore size syringe filter (Sarstedt, Nümbrecht, Germany).

### 2.2. Phage Concentration and Purification

For the concentration of the phage, ~100 mL of the filtered phage lysate (~5 × 10^9^ PFU/mL) obtained as described above (using 20 plates with confluent lysis of bacterial lawn) was used to obtain the phage pellet by centrifugation on a Beckman Optima L-100XP ultracentrifuge using a 70 Ti rotor (48,000× *g*, total centrifugation time 1 h at +4 °C; Beckman Coulter, Brea, CA, USA). The pellet was dissolved in 4 mL of TE buffer (10 mM Tris, 10 mM EDTA, pH 8.0) and the obtained suspension was layered on top of a CsCl solution (CsCl—0.6 g per mL of the TE buffer); two tubes (Ultra-Clear centrifuge tubes, 14 × 95 mm, Beckman Coulter) were filled with 11.5 mL of the CsCl solution and 2 mL of a concentrated phage sample was loaded on top of it. Tubes were centrifuged at 24,000 rpm (100,000× *g* max) for 20 h using an SW 40 Ti rotor (Beckman Coulter) at +4 °C on the Beckman Optima L-100XP ultracentrifuge. The phage-containing distinct zone was collected by pipetting and further desalted on NAP-25/Sephadex G-25 columns (Pharmacia, Uppsala, Sweden) using PBS as an exchange buffer.

### 2.3. Phage Sample Transmission Electron Microscopy and Virion Dimension Determination

The TEM visualization of the bacteriophage particles was performed using uranyl acetate negative staining of the phage-containing sample (~10^10^ PFU/mL). Particles were adsorbed from the solution to the carbonized formvar-coated 300 mesh copper grids (Agar Scientific, Stansted, United Kingdom) by a 5-min long incubation. The grids were then rinsed with 1 mM EDTA solution, stained with 0.5% uranyl acetate solution, and air-dried. The stained grids were analyzed with a JEM-1230 electron microscope (JEOL, Akishima, Japan) at an accelerating voltage of 100 kV, and the pictures were acquired with a MORADA digital camera using iTEM imaging software (Olympus, Tokyo, Japan).

The phage virion dimensions were determined using ImageJ software (v. 1.52a; [37]) with a scale bar as a reference for the pixel to nm ratio. The icosahedral capsid diameters as well as the tail lengths and widths were measured using the straight line utility provided within the software and averaged across six individual virion measurements (each parameter was measured from three individual visually intact virion micrographs taken at 300,000× magnification, and three randomly selected virions from the best micrograph taken at 60,000× magnification (multiple intact particles in the field of view)).

### 2.4. Phage Genomic DNA Extraction, Whole Genome Sequencing and De Novo Assembly

The purified phage specimen was subject to proteinase K (20 mg/mL; Thermo Fisher Scientific, Waltham, MA, USA) and SDS (0.5% final concentration; Sigma-Aldrich, St. Loui, MO, USA) treatment for one hour at +56 °C. Afterward, a Genomic DNA Clean & Concentrator-10 Kit (Zymo Research, Irvine, CA, USA) was used according to the manufacturer’s instructions to obtain the genomic DNA of the phage. The approximate concentration and purity of the obtained DNA were assessed using a NanoDrop ND—1000 spectrophotometer (Thermo Fisher Scientific) and a more precise dsDNA concentration was determined using a Qubit fluorometer (Invitrogen, Waltham, MA, USA) dsDNA high-sensitivity quantification assay (Invitrogen).

Prior to the Illumina MiSeq-compatible DNA library preparation, 200 ng of the phage DNA was fragmented using random physical shearing by a Covaris S220 focused-ultrasonicator (Covaris, Woburn, MA, USA) under the settings for a target fragment length of 550 bp. Afterward, a uniquely indexed DNA fragment library was prepared according to the TruSeq DNA Nano Low Throughput Library Prep Kit (Illumina, San Diego, CA, USA) instructions with the TruSeq DNA Single Indices Set A (Illumina). The resultant library content fitness for sequencing was assessed using an Agilent 2100 bioanalyzer (Agilent, Santa Clara, CA, USA) with a high sensitivity DNA kit (Agilent) and Qubit fluorometer (Invitrogen) dsDNA high-sensitivity quantification assay (Invitrogen). Finally, the library was sequenced as one of the 12 pooled uniquely indexed DNA libraries on the Illumina MiSeq system (Illumina) using a 500-cycle MiSeq Reagent Kit v2 nano (Illumina).

### 2.5. Phage Genome De Novo Assembly and Functional Annotation

The yield of the sequenced library and the overall quality of the raw demultiplexed reads were inspected using FastQC (v. 0.11.9; [38]). While no base quality trimming was deemed necessary prior to the de novo assembly, raw reads were pre-processed using BBDuk from the BBMap package (v. 38.69; [39]) to remove any of the potentially remaining adapter sequences and to discard reads shorter than 50 bp. Processed reads were used as an input for a de novo assembly with Unicycler (v. 0.4.8; [40]) in normal mode. A single “circular” scaffold resulting from the de novo assembly and representing a novel phage complete genome was input into PhageTerm (v. 1.0.12; [31]), along with the untrimmed reads from the respective library to check whether the physical phage genome molecule termini can be determined, and to “cut open” a circular scaffold accordingly in such a case. Afterward, the raw reads were mapped onto the “linearized” sequence of a scaffold using BWA-MEM (v. 0.7.17; [41]) and the sequence alignment map was manually inspected for any coverage dips and possible assembly ambiguities in Integrative Genomics Viewer (v. 2.4.6; [42]).

The linearized genome sequence was then imported into DNAmaster v 5.23.6 to predict open reading frames (Glimmer [43], GeneMark [44]) and possible tRNA genes (Aragorn [45], tRNAscan [46]) with the corresponding utilities within the sequence explorer. Additionally, regions between the predicted ORFs were scanned for any other putative ORFs missed during the initial step using NCBI Open Reading Frame Finder [47]. During both ORF prediction steps, only ORFs encoding a hypothetical product longer than 30 amino acids (aa) were considered, and four possible start codons were allowed (ATG, GTG, CTG, and TTG). In the case of multiple possible start codons, initial preference was given to a start codon resulting in the longest hypothetical product while not tolerating adjacent ORF overlaps that were longer than 100 bp (in such cases, the next downstream start codon was chosen).

Amino acid sequences corresponding to the putative product of each ORF were used as an input (in early November 2021) to look for conserved domains and homologous proteins: (1) NCBI conserved domain database (CDD) using NCBI conserved domain search under default settings [48]; (2) non-redundant protein sequence database restricted to viruses (taxid:10239) using BLASTp with an e-value threshold of 0.05 [49]; and (3) The Protein Data Bank (PDB), Pfam, UniProt-SwissProt-viral70, and NCBI CD databases using HHpred under default parameters [50]. The initial start codon positions were manually evaluated and corrected based on the homologous entries from the non-redundant protein sequence database (either if an experimental validation of a given homolog was published or a putatively more favorable genomic context was present in the genome of Mecenats66 for a start codon that would better match that of a homolog).

Afterward, twenty base sequences upstream of the selected putative start codons for each ORF were extracted in the 5′–3′ direction and inspected for the presence of putative Shine–Dalgarno (SD) sequences complementary to the antiSD (aSD) sequence of *Morganella morganii* (13 bases of the *M. morganii* 16S rRNA tail; 3′-AUUCCUCCACUAG-5′ [51]). For this purpose, the free_align.pl script in “helix only” (-o) mode [52] was used to identify RNA duplexes with the lowest possible change in the free energy (kcal/mol) that is required to bring the host aSD and putative SD-containing sequences upstream of the selected start codons together (allowing wobble G-U base pairing).

### 2.6. Taxonomic Identification of the Host

The DNA of the host was isolated from an overnight culture in the same fashion as described for the phage genomic DNA extraction (first paragraph the Section 2.4). Afterward, PCR (program: 3 min at +96 °C, 30 cycles of (10 s at +96 °C, 5 s at +50 °C, 4 min at +60 °C), 5 min at +70 °C, followed by a hold at +4 °C) using universal 27F and 1492R primers (ordered at Metabion, Steinkirchen, Germany) was carried out [53]. Native agarose gel electrophoresis was carried out for the PCR product, and the resulting ~1450 bp long band was extracted from the gel using the GeneJET Gel Extraction Kit (Thermo Fisher Scientific) according to the manufacturer’s guidelines.

Sanger-based sequencing [54] of the extracted 16S rRNA gene PCR product was carried out in two reactions using the aforementioned 27F and 1492R primers for the forward and reverse reads, respectively, according to the BigDye^®^ Terminator v3.1 Cycle Sequencing Kit (Applied Biosystems, Waltham, MA, USA) recommendations and using ABI PRISM 3130xl as a sequencer (Thermo Fisher Scientific). Read chromatograms were manually inspected and had their low-quality termini trimmed in GeneStudio (v. 2.2.0.0.) before being assembled into a contig, using the higher quality traces in the read overlap region in the case of any ambiguity for the consensus calling.

The resulting partial 16S rRNA gene sequence of the host was used for a comparison with the bacterial 16S rRNA gene sequences publicly available at EzBioCloud [55], The Ribosomal Database Project (RDP; [56]), and the 16S RefSeq Nucleotide sequence records [57].

For the 16S rRNA gene partial sequence phylogeny reconstruction, all valid name hits to the query sequence were retrieved from the EzBioCloud output in the form of sequences. Multiple sequence alignment (MSA) was performed using MAFFT (v. 7.453; [58]) and the resulting MSA was used for maximum-likelihood (ML) phylogeny reconstruction using IQ-TREE (v. 2.0.6; [59]) under the best fit model according to the ModelFinder [60] and allowing for polytomies; branch supports were assessed using 1000 ultrafast bootstrap (UFBoot; [61]) replicates. To visualize only high-confidence clustering, UFBoot values of less than 95% were removed from the tree using a short bash script, and the resulting tree was midpoint rooted and visualized in FigTree (Rambaut, A. FigTree v. 1.4.4. Available online: http://tree.bio.ed.ac.uk/software/figtree/ (accessed on 10 May 2021)).

### 2.7. Phage Genome Termini/Packaging Strategy Prediction In Silico

First the raw corresponding NGS library reads and the de novo assembled genome of phage Mecenats66 were used as inputs for PhageTerm (v.1.0.11.; [31]) to obtain the putative packaging strategy and genome termini prediction.

Additionally, the terminase protein amino acid sequence of phage Mecenats66 was used to infer the putative packaging strategy employed by the phage by a TerL aa sequence phylogeny reconstruction with a slightly extended dataset previously compiled by Merrill and colleagues [34]. The core dataset was extended by the addition of two sequences for an SPO1-type long direct terminal repeat clade (*Bacillus* phage BJ4 (AOZ61694) and *Bacillus* phage SBP8a (AOZ62321)) as well as the terminase from Mecenats66 (UIS74616) and two of the evolutionarily most similar proteins from other phages (determined by querying UIS74616 against non-redundant viral proteins available at the NCBI’s protein database using BLASTp; *Bordetella* phage vB_BbrM_PHB04 (YP_009792721), *Pseudomonas* phage Dolphis (QNJ57308)). Phylogeny was reconstructed using an ML approach as described in the previous subsection for host 16S rRNA ML tree reconstruction.

### 2.8. Phage Genome Termini/Packaging Strategy Elucidation In Vitro

First, the intact phage genomic DNA was used as a template for Sanger-based sequencing using custom forward and reverse primers designed to reach the putative genome 3′ and 5′ termini (Table 2).

Next, the phage genomic DNA (input ~200 ng/μL) was digested for 1 h at +37 °C using the AanI, EcoRI, EheI, SmaI, and SmiI FastDigest restrictases (Thermo Fisher Scientific), which were determined by in silico digestion to result in the digestion profiles that would present an identifiable fragment resulting from a cut near the physical beginning of the genome (putative “*pac*” fragment; see [27] for detailed elaboration on the rationale). Native agarose gel (1%) electrophoresis was performed, loading digestion reactions of the Mecenats66 genomic DNA with each of the restrictases used into individual wells and using GeneRuler 1 kb Plus DNA Ladder (Thermo Fisher Scientific) in a marker well. After visualizing the generated restriction profiles in the gel, the SmiI-generated profile was selected as the most convenient one to work with further. Thus, a larger amount of phage genomic DNA (~5 µg) was additionally digested with SmiI. Subsequently, NAGE was performed to visualize the restriction fragments and extract the fragment of interest from the gel.

The shortest (~1500 bp) faint fragment resulting from the phage Mecenats66 genomic DNA digestion with SmiI (corresponding to the putative “*pac*” fragment) was extracted from the gel using the GeneJET Gel Extraction Kit (Thermo Fisher Scientific) according to the manufacturer’s guidelines and sequenced using the *ph66_N_Rv* primer.

The Sanger read chromatograms were then aligned onto the circularized representation of the Mecenats66 genome and the junction between both genome termini was manually inspected.

All of the Sanger-based sequencing reactions for this step were prepared using a BigDye^®^ Terminator v3.1 Cycle Sequencing Kit (Applied Biosystems) and sequenced on an ABI PRISM 3130xl sequencer (Thermo Fisher Scientific).

### 2.9. Representative TerL Sequence Diversity Evaluation

The NCBI Protein database was searched for the terminase reference sequence entries of 16 March 2022 using the following query (n = 3107 returned): “(terminase [Protein Name] OR terminase large* [Protein Name] OR DNA maturation protein* [Protein Name] OR TerL [Protein Name] OR Terminase ATPase* [Protein Name]) AND (viruses[filter] AND refseq[filter])”.

Retrieved sequences were deduplicated with the help of cdhit (v. 4.8.1.; [62]) under the default parameters (sequence identity threshold of 90%), returning a set of 1046 proteins sharing less than 90% pairwise identity between them.

This dataset was then used for a conserved domain search via a web-based batch conserved domain search [63] under the default parameters. The results of the conserved domain search were then filtered on the basis of the presence of one or several of the following phage TerL-related conserved domains (n = 767 returned): pfam03237; cl26981; COG4626; pfam03354; cl12054; NF033889; COG5362; pfam05876; cl21617; COG5525; pfam04466; TIGR01547; COG1783; COG5323; COG4373; COG5565; cl11625; pfam05894.

Sequences either shorter than 200 aa in length, or sequences that were part of the dataset used by Merrill and colleagues [34] and its extension (hereafter referred to as “core TerL dataset”; see Section 2.7) were also removed.

In the end, a dataset consisting of n = 793, presumably TerL amino acid sequences (745 + 48) was used for ML phylogeny reconstruction. Multiple sequence alignment (MSA) was performed using MAFFT (v. 7.453; [43]); the resulting MSA was used for ML phylogeny reconstruction using IQ-TREE (v. 2.0.6; [44]), the best fit substitution model according to ModelFinder [45] was used; polytomies were allowed; branch supports were assessed using 1000 UFBoot [46] replicates. The resulting tree was midpoint rooted and visualized in FigTree (Rambaut, A. FigTree v. 1.4.4. Available online: http://tree.bio.ed.ac.uk/software/figtree/ (accessed on 10 May 2021)). Clades containing the most recent common ancestor of sequences representing different verified packaging strategies/genome termini types in the core TerL dataset and its children nodes are highlighted by the different colors (making an assumption that all representatives of a clade share the same packaging strategy that was also the packaging strategy MRCA used, without further more recent within-clade packaging strategy changes). Only the tips of leaves containing sequences from the core TerL dataset were assigned labels based on their packaging strategy types, corresponding leaves were also connected to the labels by colored pointed lines (black pointed lines represent leaves that were not a part of the core TerL dataset).

## 3. Results and Discussion

### 3.1. Host of Phage Mecenats66

One of the morphologically distinct colonies that formed after the incubation of the crushed-bee-derived suspension supernatant on LB agar plates was named as isolate B1-1 and further used as an indicator culture for phage screening. B1-1 was able to grow at all three of the temperatures tested (RT, +30 °C, and +37 °C), growing seemingly as fast at 30 °C and 37 °C on the LB plates. On the LB plates, B1-1 formed glistering circular colonies with an even edge that were flat and whitish-beige in color, reaching ~2 mm in diameter. The phage we named Mecenats66 was later identified using isolate B1-1 as a lawn for double-agar overlays.

Genomic DNA was extracted from an aliquot of an overnight culture of B1-1 cells and the partial 16S rRNA gene fragment was amplified and sequenced (using conventional 27F and 1492R primers). The obtained B1-1 partial 16S rRNA gene sequence revealed that different *Morganella* strain 16S rRNA gene sequences were the most similar among those available in the databases employed for taxonomic identification of the host. However, the number of nucleotide differences with the most similar ones was still noticeably high, even after the obtained Sanger read chromatograms were manually inspected for the base-call ambiguities, several of which were identified and corrected. To better understand the relationship of isolate B1-1 to other closely-related bacteria based on the 16S rRNA gene sequences, a phylogeny reconstruction approach was employed (Figure 1).

The resulting tree revealed that isolate B1-1 shared a well-supported MRCA with both of the recognized *Morganella* species representatives, and thus can be considered to belong to the genus *Morganella*. However, the evolutionary divergence of the B1-1 16S rRNA gene sequence from other *Morganella* spp. shows that it cannot be classified to the species level as of now (Figure 1). This makes further studies of this bacteria (or at least a whole genome sequencing) plausible for the possible proposal and establishment of a novel species within the genus *Morganella*, which is currently rather sparsely populated with only two species recognized as of now.

### 3.2. Morganella Phage Mecenats66 Virion Morphology and Genome Overview

*Morganella* phage Mecenats66 virions have a myovirus morphology: icosahedral heads with a diameter of 75.6 ± 3.9 nm to which 142.1 ± 3.7 nm long and 19 ± 1.7 nm wide contractile tails ending with a lush basal plate are attached (Figure 2, Appendix A). 

Sequencing of the Mecenats66 genomic DNA library resulted in 92,029 read pairs of up to 251 bp in length, which were preprocessed and de novo assembled into an 86,193 bp long “circularized” DNA sequence with a 49.07% GC content representing the complete genome of phage Mecenats66, which had a mean depth of 514× and each base was covered at least 292×. PhageTerm analysis using raw corresponding library reads resulted in the identification of a base having a read starting position coverage considerably higher than all other bases, which was predictive of a headful packaging strategy with a preferred packaging initiation site, implying obvious termini, and the de novo assembled genome was reorganized accordingly.

Open reading frame prediction resulted in the documentation of 123 ORFs and revealed a coding capacity of ~96%. Functions for putative products encoded for 84 (~68.3%) out of the 123 ORFs could not be reliably predicted using the annotation methodology employed. Moreover, 52 putative products had no BLASTp hits to proteins encoded by other publicly available phage genomes, and six more had hits with an expect value higher than the selected confidence threshold used for taking the hits into account for annotation (≥1 × 10^−3^), indicating that Mecenats66 is quite distinct from the currently uncovered phage diversity. ATG was chosen as a start codon for 115 of the ORFs, GTG for seven, TTG for a single ORF, and none of the ORFs were assumed to have a CTG start codon (Appendix A).

When queried against all of the publicly available viral entries using BLASTN (taxid:10239), no hits with a query coverage to the *Morganella* phage Mecenats66 complete genome greater than 3% were documented, implying an immense intergenomic distance of greater than 97% to any of the so far sequenced phages whose genomes are publicly available. This by far exceeds the ICTV BVS set criteria for novel phage species (95% intergenomic similarity to taxonomically recognized phage) and genus (~70% intergenomic similarity to taxonomically recognized phage genera representatives) level demarcation criteria [64], as the intergenomic similarity of Mecenats66 to any of the phage genomes elucidated as of now does not exceed 3%.

Based on the origin of the top-scoring Mecenats66 protein amino acid sequence BLASTP hits, however, *Pseudomonas* phage Dolphis and *Bordetella* phage vB_BbrM_PHB04 were the only phages whose proteins were the most similar to their counterparts in Mecenats66 in the case of more than three individual proteins. In this way, homologs from Dolphis were the highest-scoring hits for 27 of the Mecenats66 proteins, while homologs from vB_BbrM_PHB04 were seen as the highest-scoring for 13 of the Mecenats66 proteins (Appendix A). A comparison of the genome organization and protein homology of these phages to Mecenats66 revealed that their similarity was largely restricted to the proteins involved in phage particle morphogenesis or virion structural components. All three also had some homologous proteins presumably involved in DNA replication, modification, and repair (in Mecenats66: helicase encoded by ORF99, an ssDNA-binding protein (ORF100), putative nuclease (ORF108), and DNA primase/helicase (ORF109)), an ATPase (ORF96), and a hypothetical protein encoded by ORF95 of Mecenats66 (Figure 3). Given their low pairwise genome nucleotide sequence similarity, it is not unexpected that the proteome contents of these three phages differed greatly, although these differences are largely confined to the hypothetical proteins that did not have any functional assignment (Appendix A).

As we were not able to locate both terminase subunits (TerS and TerL), but predicted that ORF56 in the Mecenats66 genome encoded a product of 873 amino acids in length (UIS74616.1) that had an identifiable “Phage terminase large subunit (GpA)” conserved domain (cl21617; Pssm-ID: 389842; conserved domain length: 559; E-value: 9.85 × 10^−17^), we opted to label it as just a “terminase”, as is common in such cases.

### 3.3. Exact Genome Termini and Genome Packaging Strategy Elucidation for Mecenats66

A widely used terminase large subunit amino acid sequence-based packaging strategy prediction using a similar approach and the extended dataset of Merrill and colleagues [34] did not support a headful packaging strategy for Mecenats66 and two similar proteins of comparable length from other cultured and sequenced phage proteomes (*Pseudomonas* phage Dolphis and *Bordetella* phage vB_BbrM_PHB04). Terminase/TerL sequences of these three phages formed a distinct and well-supported clade on their own, which confidently clustered together with 3′ cos (Lambda) and the long direct terminal repeat (LDTR; SPO1) genome termini type/packaging strategy employing phage clades. The evolutionary distances between Mecenats66 and both Dolphis and vB_BbrM_PHB04 suggest that all three of these phages might employ the same packaging strategy (Figure 4).

As the exact termini elucidation has not been performed for *Pseudomonas* phage Dolphis (MT711888.1; unpublished) or *Bordetella* phage vB_BbrM_PHB04 (MF663786.1; [66]) to the best of our knowledge, a decision to experimentally verify the exact physical termini of phage Mecenats66 was made, with a possibility of extending the results to both of these phages, whose terminase sequences share an MRCA with their counterpart from Mecenats66 (Figure 4).

For this, the PhageTerm reorganized genomic nucleotide sequence of Mecenats66 was used for virtual restriction enzyme digestion analysis to select restrictases that would cut the molecule close to the predicted rightward termini (packaging series initiation site-containing/*pac* fragment) and would overall result in a digestion profile that would not be “overcrowded” by fragments of lengths similar to the generated *pac* fragment which would hamper its unambiguous identification. Eventually, five restrictases that met these criteria were chosen (restrictase—*pac* fragment size): AanI—1581 bp, EcoRI—754 bp, EheI—213 bp, SmaI—7219 bp, SmiI—1434 bp). Upon performing the experimental restriction, however, weak bands corresponding to the expected length *pac* fragments were identifiable in NAG only for three of the restrictases used (Figure 5, Appendix A). It turned out that SmaI was not able to digest the Mecenats66 genomic DNA at all, and no fragment of expected length was seen for EheI. The expected EheI-generated *pac* fragment might just not be visible due to its expected short length (213 bp), which could result in the amount of ethidium bromide molecules associated with it being lower than necessary to emit visible fluorescence at the input DNA amount used for restriction. The complete absence of any fragments cut from the genomic DNA (while 36 of them were expected) in the case of SmaI might be indicative of extensive methylation of the phage Mecenats66 genome as SmaI completely overlaps CpG and its activity is thus blocked by CpG methylation. Out of the digestions with restrictases that presented a *pac* fragment, SmiI was chosen to further digest a higher amount of Mecenats66 genomic DNA (~5 µg in total), excise the *pac* fragment from the gel, and sequence it.

Reads from the intact genomic DNA of Mecenats66 sequencing using primers *ph66_C_Fw* and *ph66_N_Rv* showed that sequences spanned over the predicted termini, implying that the genome of Mecenats66 is terminally redundant, indicative of headful packaging (Figure 6). Bases at the termini junction were called unambiguously in the case of *ph66_C_Fw*; the read originating from sequencing using *ph66_N_Rv* demonstrated ambiguity at the last base of the genome (position 86193), with the de novo assembled base guanine at the given position (also seen in case of sequencing with *ph66_C_Fw*) having an approximately thrice as less intense peak corresponding to thymine in reverse complement of the corresponding read. Given that the reverse-complement of the sequence obtained from *ph66_N_Rv* is presented in Figure 6 (tile 2), the original read had a minor peak called adenine, which is an expected non-template single-strand overhang that is added to the physical end of a DNA molecule being sequenced using the polymerase employed. This allowed us to hypothesize that the phage Mecenats66 genome concatemer formed during the replication indeed has a preferred packaging initiation site, which was predicted correctly using the PhageTerm approach (Figure 6). This was further validated when a SmiI-generated pac fragment was sequenced using the *ph66_N_Rv* primer. In this case, the intensities of the base call peaks suddenly dropped beyond the −1 base position, representing a non-specific single base overhang added by the polymerase. The peaks upstream of the presumed start site, however, coincided with the leftward terminus sequence of the Mecenats66 genome, although it had nearly invisible peak intensities when compared to the rightward terminus sequence. This further confirmed the presence of the preferred packaging series initiation site, which was correctly identified from the starting NGS read position coverage comparisons in silico. The low-peak-intensity sequence before it, however, additionally validated that Mecenats66 formed concatemers during its genome replication, and these concatemers were packed into the procapsids in a headful fashion, where the DNA molecule cuts were made in a non-specific manner when the procapsid was physically “full”. Although we were not able to determine the extent of the terminal redundancy of Mecenats66 genomes and/or estimate the genome copy numbers within the concatemers of the Mecenats66 genome during replication, the aforementioned experimental validation confirmed that Mecenats66, indeed, employs a headful genome packaging strategy with a preferred packaging initiation site, as correctly predicted by PhageTerm from the raw reads.

We also additionally sequenced another independent batch of genomic Mecenats66 DNA using the same NGS library preparation protocol and sequencing technique as the first time, with a modification that the second time, 100 ng of the Mecenats66 genomic DNA was pooled with 100 ng of DNA from another completely unrelated phage under a single adapter. The resulting Mecenats66 genome assembly was also circular, but as expected, circularly permuted to the first one. PhageTerm analysis using the second sequencing run raw reads and the second Mecenats66 de novo assembled genome, however, resulted in the reorganization of the genome precisely at the same genome starting position as in the initial sequencing run case (44% of the pooled two distinct phage library reads could be mapped unto the assembled genome in the second sequencing). This second reorganized assembly, which had a mean sequencing depth of 227× and a whole genome depth of 77× (none of the bases were covered by fewer individual reads) also had a very similar base coverage histogram to the initial reorganized assembly (Appendix A), albeit, understandably, with more than a two-fold lower mean depth (although not very different when a correction on a given library read percentage mapped onto the assembly is taken into account). Of interest, whereas the same headful packaging strategy with the same preferred packaging series initiation site was also determined by PhageTerm from the data independently generated in the repeated sequencing experiment using a new library preparation, the concatemer estimation PhageTerm produced was different. Based on the initial data, a 5.0 concatemer size was estimated, whereas the repeated sequencing data resulted in a concatemer size estimation of 3.0, although this incongruency may be a result of different read datasets used as an input (initially a single phage library with 96% reads mapped onto the assembled Mecenats66 genome, repeatedly—two different phages pooled within a single library with 44% of the reads mapping unto the assembled Mecenats66 genome), and its origin was not studied further. The ratio of the highest read starting position coverage to the second highest read starting position coverage on the same strand, however, also differed between the inputs, being ~9 in the case of the initial sequenced library (52/6 = 8.67) and ~7 (34/5 = 6.80) in the case of repeated library prep and sequencing. Thus, these estimates might indeed be more precise with the increasing depth of each sequenced base in the genome (e.g., PhageTerm prediction using the method by Li and colleagues [32] shows a warning if the whole genome coverage is less than 200×).

### 3.4. Robustness of the TerL/Terminase Amino Acid Sequence Phylogeny Reconstructions to Predict the Packaging Strategy of a Novel Phage

Validation of the headful packaging strategy with a preferred packaging series initiation site for the *Morganella* phage Mecenats66, however, has prompted us to explore the representative currently available TerL/terminase amino acid sequence diversity and check the extent of usability of a previously proposed set of sequences from phages with verified genome termini/packaging strategies [34] for the prediction of the packaging strategies of novel phages. Thus, to test the robustness of a TerL aa sequence phylogeny reconstruction-based approach and visualize the diversity of representative phage TerL aa sequences as well as their clustering, the ML tree was built for nearly 800 divergent TerL aa sequences from phages as described in the Materials and Methods Section 2.9 (Appendix A, Figure 7).

The resulting tree was used to evaluate the extent of the TerL/terminase aa sequence clustering with a subset of sequences originating from phages for which the packaging strategy/genome termini were verified experimentally (“core TerL dataset”). Well-supported clades of sequences corresponding to the core TerL dataset packaging strategy groups were highlighted in different colors (based on the extension from the last MRCA of each packaging group sequences within a wider context than that presented in Figure 4). Although the employed approach to packaging strategy designation might be considered somewhat conservative (nevertheless having its own assumptions; Section 2.9), as most of the analyzed sequences remained without a prospective packaging strategy assignment, the extent to which a terminase phylogeny-based approach can explain the packaging strategy/termini type employed by a novel phage remains unclear for phages having sufficiently distinct TerL/terminase amino acid sequences from the ones for which previous experimental verification has been carried out. As can be seen from Figure 7, some of the well-supported but evolutionary very distinct clades from the sequences within the core TerL dataset sequence clades are evident. Moreover, sequences from two of the packaging strategy groups from the core TerL dataset seemed to break when a considerably more diverse homologous sequence context was presented for an alignment. This was especially evident for the phages that presumably employed the Host End (e.g., Mu) packaging strategy, while two of the corresponding core TerL dataset sequences (from the *Burkholderia* phage BcepMu and *Escherichia* phage Mu) clustered together to share a well-supported MRCA, one (from *Pseudomonas* phage B3) has been predicted to have completely different ancestry, although deep branching is unreliable given the diversity of sequences presented for an MSA from which the tree was generated. To a lesser extent, this was also the case for the SDTR (c-st) phages, where sequences from *Clostridium* phage c-st and *Bacillus* phage Basilisk shared a well-supported common ancestor between themselves, but were quite far away from the terminase sequence of *Bacillus* phage vB_BanS_Tsamsa, despite all of them being classified as the c-st type SDTR employing phages. Sequences representing other packaging strategy groups, however, behaved as would be expected from a core TerL dataset only, and had their respective well-supported MRCAs. Although most of the sequences in the tree did not fall within any of these groups, the sequences that did, however, presented quite the differences in evolutionary conservation (pairwise lengths of branches connecting tips falling within particular highlighted clades) among its members. Notably, only the phages found in RefSeq were used for the generation of a “representative” dataset herein, however, overall, there are way more phage complete genomes available publicly elsewhere in the INSDC databases, and one would expect that at least a fraction of them would give rise to more distinct clades, thus making the relationships between different packaging strategies employing phage terminases/TerL sequences even more complicated to be interpreted as of now.

This, sadly, stems from a recurrent problem that most of the bacteriophages, whose complete genomes are available at public biological sequence repositories, do not have their exact genome termini/packaging strategy elucidated and/or indicated, which makes trying to understand the extent of pairwise TerL evolutionary distances ensuring the retention of the same packaging strategy difficult. The phylogeny of a “representative” TerL diversity dataset showed an intriguing clustering pattern of the sequences analyzed, indicating that a widely used dataset consisting of terminase sequences from phages that have had their packaging strategy experimentally verified, although useful, encompasses only a small fraction of this diversity (Figure 4 vs. Figure 7).

As there seem to be clusters of sequences showing plausible MRCAs that seem to be evolutionarily quite distant from any of the experimentally verified packaging strategy TerL/terminase sequence-containing clades, elucidation of the genome termini, at least for some of the phages falling within those “unknown packaging strategy” clades, might further refine the applicability of TerL-phylogeny based approaches for the packaging strategy prediction of novel phages. Most importantly, however, additional data on this subject could also shed light on the evolution of different dsDNA-containing phage packaging strategies per se: was the arguably least specific headful packaging strategy the initial one, from which all the others had evolved; how many times did these different packaging strategies evolve; which are the molecular constraints of the TerL aa sequence for a particular packaging strategy, etc.? These global questions are currently far from being answered and would undeniably benefit from more phage genomes having their physical termini/packaging strategy determined. Furthermore, the disregard for the exact genome termini/packaging strategy elucidation that is seen in many of the publicly available phage genomes has led to the accumulation of complete phage genomes whose genome sequence representation is arbitrary, and probably depicts a scaffold “as spat out by the de novo assembler employed”. The lack of efforts of such submission authors is especially worrisome when their submitted phage genome is very closely related to some of the most well-studied ones (e.g., a single copy of the SDTR region in the middle of a genome for a T7-like phage whose sequence is nearly identical to the one in phage T7), which makes organizing the genome sequence being submitted at least somewhat (or even completely in some cases) representative of a physical phage genome molecule “as seen within the capsid” possible. The process might take just as much time as necessary to add a single line of code evoking PhageTerm (the results of which usually coincide with the further experimental verification) as a step in the pipeline after the de novo assembly, even if no TerL phylogenies/comparative analyses for trying to organize the newly sequenced phage genome correctly are to be performed. The concept behind PhageTerm, however, as highlighted by its authors [31], requires the random fragmentation and availability of the termini for adapter ligation (e.g., random physical shearing using sonication and Illumina’s TruSeq kits would result in a read dataset that would be usable, whereas Illumina’s transposon-based Nextera kits would result in a read dataset for which the PhageTerm result would be meaningless for termini elucidation).

## 4. Conclusions

Novel *Morganella*-infecting myovirus Mecenats66 isolated from dead worker honeybees demonstrated a spectacular genome nucleotide sequence uniqueness within the thus far uncovered phage diversity. Mecenats66 seems to be the eighth completely sequenced *Morganella*-infecting phage, and the first among them to be isolated from an insect-associated source, along with its host bacteria strain, which might represent a novel *Morganella* species based on the nearly full-length 16S rRNA gene sequence.

Genome physical termini scrutinization allowed us to conclude that Mecenats66 employs an evolutionary distinct subtype of headful packaging with a preferred packaging series initiation site. A detailed workflow describing three of the approaches (NGS read pile-up inspection; TerL/terminase phylogeny reconstruction; Sanger-based sequencing of genome termini from the intact genomic DNA and terminus-containing genome restriction fragment) that were used to predict/validate the packaging strategy is presented.

Although very often overlooked, experimental validation of the packaging strategy/genome termini type of newly isolated and sequenced phages is of importance to improve the understanding of packaging strategy evolution among the dsDNA-containing phages and is highly encouraged. Based on the observed representative TerL/terminase amino acid sequence clustering, elucidation of the termini for at least a single phage that convincingly falls within one of the well-supported TerL clusters can most likely be extended to all members of the cluster, although this is somewhat subjective at this point as the extent of evolutionary distances and TerL sequence constraints that allow for the retention of the same packaging strategy is not currently clear.

## Figures and Tables

**Figure 1 microorganisms-10-01799-f001:**
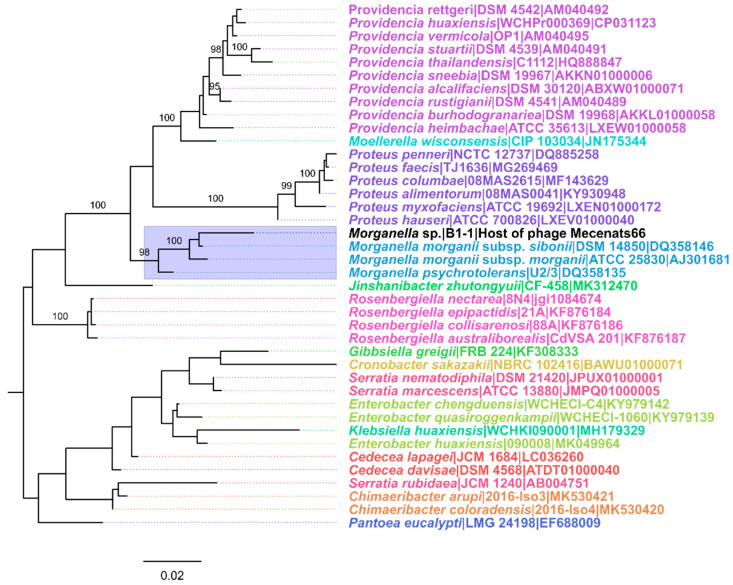
The maximum-likelihood tree of the partial 16S rRNA gene sequences of the bacterial isolate B1-1 and closely-related bacterial species. The analysis involved 40 nucleotide sequences (16S rRNA gene sequences of isolate B1-1 and 39 other most closely related bacterial species identified). The tip labels correspond to the taxa and are in the format of “Species|Strain|Accession” and are colored based on the genus of the bacteria from which the sequence was derived. Isolate B1-1 is indicated by the black font, and the clade corresponding to the 16S rRNA sequences of bacterial genus *Morganella* representatives is highlighted in light blue. Input alignment had 1465 columns, 284 distinct patterns, 183 parsimony-informative, 41 singleton sites, and 1241 constant sites. The tree was built using TN + F + I + G4 as the best-fit substitution model. Near zero-length branches were collapsed into polytomies. The tree shown is midpoint rooted. The percentage of replicate trees in which the associated sequences clustered together in the ultrafast bootstrap (UFBoot; 1000 replicates) is shown next to the branches for branches with UFBoot support higher or equal to 95%. The tree is drawn to scale, branch lengths represent the number of nucleotide substitutions per site.

**Figure 2 microorganisms-10-01799-f002:**
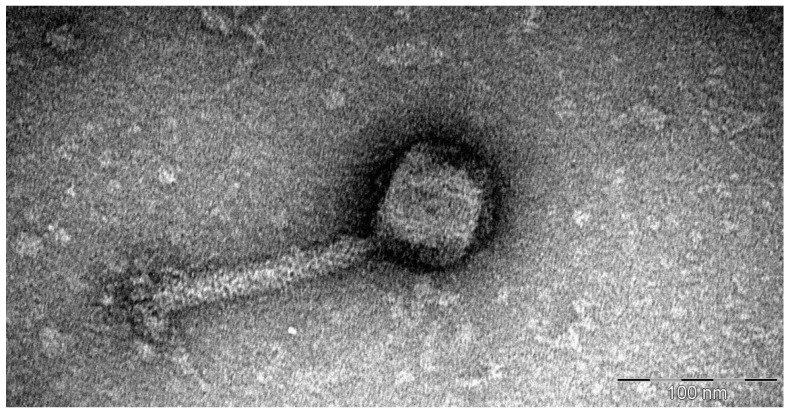
The transmission electron micrograph demonstrating the morphology of the *Morganella* bacteriophage Mecenats66 virion negatively stained with 0.5% uranyl acetate.

**Figure 3 microorganisms-10-01799-f003:**
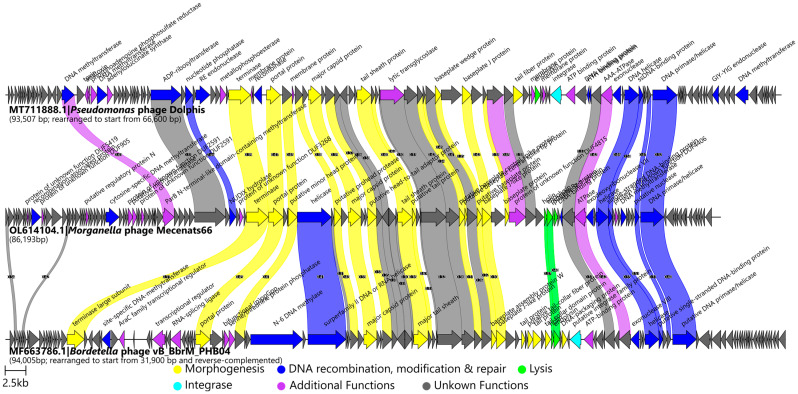
Pairwise genome organization and encoded protein comparison of *Morganella* phage Mecenats66 (**middle**) to *Pseudomonas* phage Dolphis (**upper**) and to *Bordetella* phage vB_BbrM_PHB04 (**lower**). Genomes are drawn to scale, and the genomes of Dolphis and PHB04 were reordered as indicated to ensure collinearity with Mecenats66; the scale bar indicates 2500 base pairs. Arrows representing open reading frames point in the direction of the transcription and are color-coded based on the function of their putative product according to the legend. Slanted labels above the arrows indicate the predicted function for the given ORF putative product in the case it had a function assigned (original annotations were retained for Dolphis and PHB04; predicted Mecenats66 ORF123 crossing the genome termini junction is not shown). Ribbons connect homologous proteins of phages and are colored according to the predicted functional group of the respective Mecenats66 ORF product. Ribbon labels indicate the pairwise similarity between the connected ORF products (for products having ≥30% identity). Comparisons were carried out using clinker (v.0.0.23.; [65]).

**Figure 4 microorganisms-10-01799-f004:**
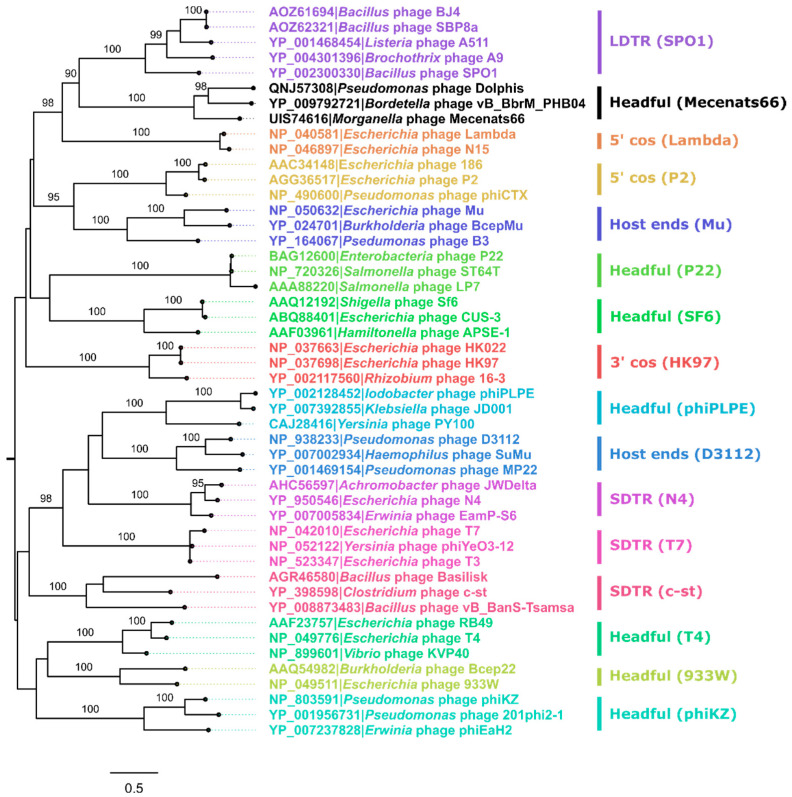
The maximum-likelihood tree of terminase/terminase large subunit (TerL) amino acid sequences for the prediction of the phage Mecenats66 packaging strategy. Input alignment had 48 sequences with 1539 columns, 1482 distinct patterns, 996 parsimony-informative, 346 singleton sites, and 197 constant sites. The tree was built using VT + F + R4 as the best-fit substitution model. Near zero-length branches were collapsed into polytomies. The tree shown is midpoint rooted. Tip labels are colored based on the distinct packaging strategies the phages seen in the tree employ (experimentally verified for most of the phages represented in the tree [34]). The percentage of replicate trees in which the associated sequences clustered together in the ultrafast bootstrap (UFBoot; 1000 replicates) is shown next to the branches for branches having a UFBoot support higher or equal to 95% (except for the MRCA node of the SPO1 type and Mecenats66 type TerL/terminase sequence clades for which a UFBoot of 90% is also shown). The tree is drawn to scale, and branch lengths represent the number of amino acid substitutions per site. Tip labels correspond to the phages from which the respective terminase/TerL amino acid sequences were derived and are in the format of “Protein accession|Phage”. Colored bars next to the labels indicate evolutionary distinct TerL clades, which correspond to different packaging strategies (LDTR stands for long direct terminal repeats, SDTR—short direct terminal repeats).

**Figure 5 microorganisms-10-01799-f005:**
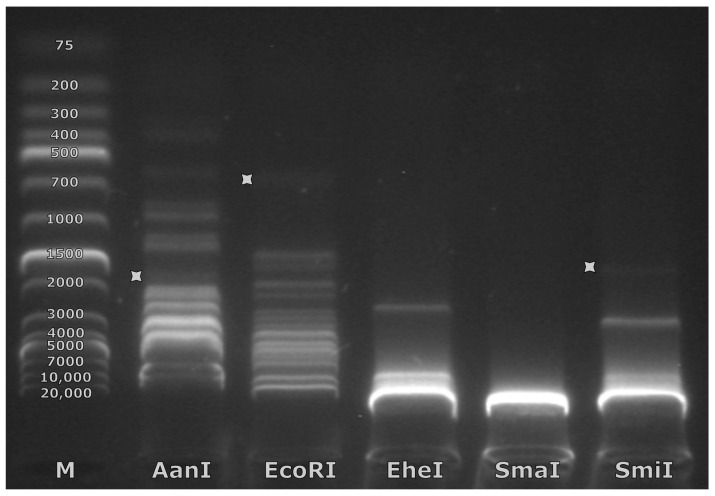
Restriction enzyme generated fragments from *Morganella* phage Mecenats66 DNA digestion by the selected restrictases. Tracks are named according to the FastDigest restrictase used. Track marked “M” indicates a marker/ladder track, and numeric values on the fragments indicate the ladder fragment sizes in base pairs. In the tracks named AanI, EcoRI, and SmiI, faint bands corresponding to the *pac* fragments are indicated by grey stars to the right of the respective fragment.

**Figure 6 microorganisms-10-01799-f006:**
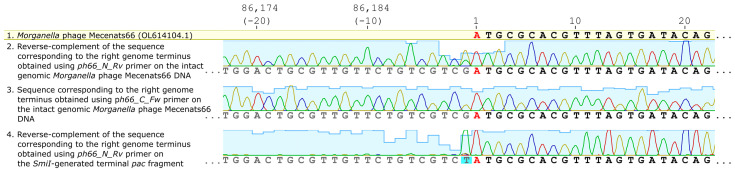
The *Morganella* phage Mecenats66 exact genome termini verification. The first track shows the rightward terminus nucleotide sequence of the Mecenats66 genome (first 22 bases). Tracks 2, 3, and 4 show Sanger-based sequencing read chromatograms corresponding to both termini, peaks are colored based on the base-call (red—adenine, green—thymine, yellow—guanine, blue—cytosine) and their heights represent the relative signal intensities. Cyan bars in the background represent the relative quality of the called base. In all of the tracks, the first base in the genome of Mecenats66 corresponding to an adenine is marked in red. In track 4, the thymine base with a cyan background (an adenine in the non-reverse-complemented original read) represents a non-template single base overhang added after the physical end of the most sequenced *pac* site-containing molecules.

**Figure 7 microorganisms-10-01799-f007:**
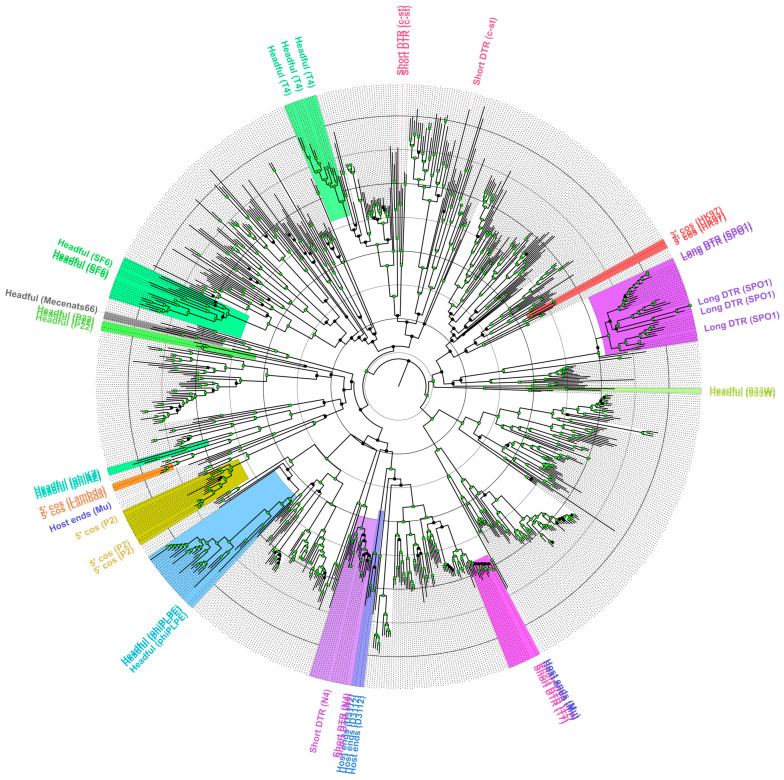
The maximum-likelihood tree of representative terminase/terminase large subunit (TerL) amino acid sequence diversity. Input alignment had 793 sequences with 6470 columns, 4725 distinct patterns, 3011 parsimony-informative, 1119 singleton sites, and 2340 constant sites. The tree was built using Blosum62 + R10 as the best-fit substitution model. Near zero-length branches were collapsed into polytomies. The tree shown is midpoint rooted. Tip labels are provided for phages that had their packaging strategy/genome termini type verified experimentally and are colored based on the distinct packaging strategies phages within the presumed same packaging strategy clades employ (sequences from the extended “core” dataset have their respective packaging strategies indicated at the corresponding tips [34]; Figure 4) The clade consisting of the terminase sequence from Mecenats66, Dolphis, and PHB04 is highlighted in grey. LDTR stands for long direct terminal repeats, SDTR for short direct terminal repeats. Distal nodes of branches having UFBoot support higher or equal to 95% are indicated by green squares. The tree is drawn to scale, branch lengths represent the number of amino acid substitutions per site, and the radial circles are spaced 0.5 amino acid substitutions per site apart.

**Table 1 microorganisms-10-01799-t001:** The known *Morganella*-infecting bacteriophages. Family and genus columns are based on genome accession associated taxonomy. Asterisk (*) indicates viral families recently abolished by the ICTV, but either still present in the associated genome accession taxonomy at the time of writing, or indicated in the reference for phage FSP1 for which no complete genome is yet available.

Genome Accession	Phage	Genome Length (kbp)	GC% Content	CDS Number	Family	Genus	Isolation Source	Host	Reference
EU652770	*Morganella* phage MmP1	38.457	46.52	49	*Autographiviridae*	*Minipunavirus*	Sewage	*Morganella morganii*	[11]
KX078568	*Morganella* phage vB_MmoP_MP2	39.616	46.91	55	*Autographiviridae*	*Minipunavirus*	Sewage	*Morganella* sp.	[12]
KX078569	*Morganella* phage vB_MmoM_MP1	163.201	34.75	271	*Straboviridae*	*Gualtarvirus*	Sewage	*Morganella* sp.	[12]
KY653118	*Morganella* phage IME1369_01	47.739	45.82	67	*Siphoviridae **	not defined	Host (prophage)	*Morganella morganii* IME1369	N/A
KY653119	*Morganella* phage IME1369_02	39.387	50.02	60	*Siphoviridae **	not defined	Host (prophage)	*Morganella morganii* IME1369	N/A
OK499982	*Morganella* phage vB_MmoP_Lilpapawes	39.168	47.06	57	*Autographiviridae*	*Minipunavirus*	Sewage	*Morganella morganii*	[13]
OK499989	*Morganella* phage vB_MmoM_Rgz1	165.808	34.71	275	*Straboviridae*	*Gualtarvirus*	Sewage	*Morganella morganii*	[13]
OL614104	*Morganella* phage Mecenats66	86.193	49.07	123	*Myoviridae **	not defined	Insect gut	*Morganella* sp.	This study
N/A	*Morganella* phage FSP1	~45.6–49.4	N/A	N/A	*Myoviridae **	N/A	River water	*Morganella morganii* ssp. *morganii*	[14]

**Table 2 microorganisms-10-01799-t002:** Custom primers designed to elucidate the physical genome molecule termini of the phage Mecenats66.

Primer	Primer Sequence (5’–3’)	Coordinates in Mecenats66 Genome (Bases)
*ph66_C_Fw*	GCAGCAGGATCGTTAAGTCC	85,939–85,958
*ph66_N_Rv*	CTAGATCGCATCGATTGCAG	334–353

## Data Availability

The annotated complete genome sequence of *Morganella* phage Mecenats66 reported herein is available at GenBank under accession OL614104.1. The accession numbers of the other phage genomes or their terminase/TerL amino acid sequences used in the study are listed in either the figures/tables or Appendix A.

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
