# Peer review of "Morganella Phage Mecenats66 Utilizes an Evolutionarily Distinct Subtype of Headful Genome Packaging with a Preferred Packaging Initiation Site"

_microorganisms, 2022, doi:10.3390/microorganisms10091799_

Round 1

Reviewer 1 Report

The manuscript "Morganella phage Mecenats66 utilizes an evolutionarily dis- 2 tinct subtype of headful genome packaging with a preferred 3 packaging initiation site" by Zrelovs et al describes the genomic characterization of an insect associated Morganella phage isolated from the dead worker honeybees. bioinformatic and wetlab approaches suggest that the isolated phage utilizes and evolutionarily distinct headful genome packaging strategy with a previously unrecognized  preferred packaging initiation site strategy subtype. The manuscript is reasonably well written, bioinformatic and experimental approaches seem to be well considered, and the overall topic and findings might be of interest to a specialized journal like microorganisms.

That said, the manuscript could be improved if the authors could provide some mechanistic insight into the use of TerL proteins to distinguish between packaging sub-types. For example, monomeric TerL structures have been determined for several headful-packaging phages and unit-lenght packaging phages. Further structures of functional rings have now been published, along with comprehensive translocation mechanisms. Considerable mechanistic insight has thus been extracted from these structures. Hence, what aspects of sequence suggest a packaging sub-type. Does the location of the corresponding amino acids suggest any mechanistic feature that would help with the distinction between packaging subtypes, or is the entire analysis depend simply on correspondence? Similarly, detecting specific start sights usually depends on TerS - is there any information to be gleaned from examining TerS sequences?

Author Response

Reviewer 1: The manuscript "Morganella phage Mecenats66 utilizes an evolutionarily distinct subtype of headful genome packaging with a preferred packaging initiation site" by Zrelovs et al describes the genomic characterization of an insect associated Morganella phage isolated from the dead worker honeybees. bioinformatic and wetlab approaches suggest that the isolated phage utilizes and evolutionarily distinct headful genome packaging strategy with a previously unrecognized  preferred packaging initiation site strategy subtype. The manuscript is reasonably well written, bioinformatic and experimental approaches seem to be well considered, and the overall topic and findings might be of interest to a specialized journal like microorganisms.

That said, the manuscript could be improved if the authors could provide some mechanistic insight into the use of TerL proteins to distinguish between packaging sub-types. For example, monomeric TerL structures have been determined for several headful-packaging phages and unit-lenght packaging phages. Further structures of functional rings have now been published, along with comprehensive translocation mechanisms. Considerable mechanistic insight has thus been extracted from these structures. Hence, what aspects of sequence suggest a packaging sub-type. Does the location of the corresponding amino acids suggest any mechanistic feature that would help with the distinction between packaging subtypes, or is the entire analysis depend simply on correspondence? Similarly, detecting specific start sights usually depends on TerS - is there any information to be gleaned from examining TerS sequences?

Author reply:

We thank Reviewer 1 for a very interesting suggestion! While additions like that could, indeed, be of interest to the readership of microorganisms, we, sadly, do not see any possibility of including such data within this particular manuscript while doing this suggestion justice. There are several reasons why we think it would not be a good idea to approach this topic herein. Mainly, while undeniably very connected to the topic of our manuscript, the number of additions and extra in silico work necessary to adequately approach this suggestion will greatly enlengthen and complicate the current manuscript, as we see it, and will push an additional “third” dimension into it. In our opinion this suggestion warrants a possibility for a completely another fully fledged article altogether that would in silico explore the possibilities of outlining/observing conserved structural and functional features, and how are they supported by the corresponding packaging strategy sequence profiles to try to draw conclusions about the importance of residues for packaging strategy distinction.

Obviously, the addition of structural TerL/terminase background would greatly enlengthen the introduction that already is composed of two somewhat stand-alone parts that were hard to glue together. We oppose “salami slicing” in publishing, and we did not want to first publish just an article on genomic characterization of Mecenats66, and then another article on its packaging strategy-related work and its place within a broader context, thus, these two topics are already present in this manuscript. The addition of a third one, actually, as we envisage the implementation of this suggestion, would have no real connection to Mecenats66 itself and would focus exclusively on a large dataset and its part analyses. With this, we are brought to the fact that actually, an analysis to answer the interesting questions you raised would have to be done from scratch, and would be, without exaggeration, massive. The current representative TerL/terminase diversity MSA that was used, we think, might be very unreliable for comparisons like that (e.g. number of columns (6470) severalfold surpasses the longest sequence used as input for the alignment; only 409/6470 columns contain less than 50% gaps). Moreover, we think that approaching questions like that would better benefit from looking at a more redundant dataset (e.g. not deduplicating at 90% pairwise sequence identity), if, theoretically, just a few amino acids might be of importance. Given the distances between sequences in the tree, where some immense sequence divergence might be observed, we cannot trust the deeper branching much, regardless of the branch supports, thus, a tree is interpreted rather cautiously, focusing on the shallow external branching. Actually, the clades of the putatively ”same packaging strategy employing phages” in a larger sequence diversity context tree were also highlighted based on an assumption that the latest well-supported MRCA of the sequences that come from phages that had packaging experimentally verified has had the same packaging strategy, which was extended to its other descendants. This assumption may as well be wrong, as, theoretically, some of the leaves within these monophyletic packaging strategy clades may have diverged to change their packaging strategy but lack experimental verification. Thus, for such an analysis, first of all, compilation of a dataset having more terminases/TerLs from many more phages that had their packaging strategy verified would be needed to draw better conclusions and inform the work. If we ignore the redundancy, we believe that phages sequenced and characterized within the SEA-PHAGES program (e.g. https://phagesdb.org) might greatly aid in the creation of such a dataset, as people involved in the program tend to elucidate the genome termini of their phages. Next, after performing a yet again very likely not a very qualitative MSA given the diversity of sequences that would be analyzed, a tree would have to be built to check for clades including phages with experimentally verified packaging strategy and other children for all the presumably same packaging strategy cluster phages. Afterward, MSAs should be performed for each presumed packaging strategy cluster (this would greatly reduce the pairwise distances between the sequences being aligned and, thus, would be expected to provide a much better and reliable alignment of within-cluster sequences), and sequence profiles ought to be generated. These steps might be needed to be repeated several times for several clusters. Profiles of presumably different packaging strategy employing phage terminases could then be compared to point out the key differences between the amino acids at given positions. This might as well allow spotting within cluster evolution of presumably different packaging strategies that differed from MRCA of the sequences that had the same packaging strategy experimentally verified. This, however, would also have to include previously experimentally obtained protein structure analysis with the results of sequence profiles in mind and would involve closer collaboration with some structural biologists working in this area, as we do not have any of the co-authors with a sufficient structural biology experience in this particular subject.

While, like most things, the interesting suggestion you proposed might probably be approached to be implemented in a different way, this is what we could come up with. We believe that our elaboration on this shows, that partaking in such an endeavor would be far from a trivial task, and the amount of data generated would be hefty enough to warrant a rather lengthy and complicated manuscript of its own. While the suggestion is very interesting and is being considered by us, we see no possibility of trying to do this within the frame of this manuscript as it would likely grow twice its current size and get overly complicated, while losing connection to phage Mecenats66 itself and its packaging strategy, which was a cornerstone herein. We believe that this manuscript, in case of publishing, might attract correspondence from like-minded individuals with whom we could possibly collaborate in the future to try taking on the studies on evolutionary aspects of phage packaging in an in-depth manner (additionally, currently, we lack relevant structural biology expertise to successfully partake such a study on our own).

Thus, indeed, speaking of global context, our analysis performed herein allowed us to conclude that Mecenats66 uses an “evolutionarily distinct subtype” of headful genome packaging with a preferred packaging initiation site (shown experimentally) depends on a lack of correspondence of Mecenats66 terminase to any of the other phage, that had their packaging strategies/genome termini experimentally verified terminases/TerLs without dwelling deeper into the sequence/protein structure evolutionary aspect of phage packaging strategies in a very in-depth manner as per your suggestion.

Speaking of TerS, the point is that the ORF encoding it was not unambiguously identifiable in the genome of Mecenats66. Although we are perfectly aware that two ORFs encoding, respectively, TerS TerL subunits, are usually found in phage genomes (often in close proximity), we have failed to identify a potential TerS-encoding ORF candidate in the genome of Mecenats66. TerS sequences are not as evolutionarily conserved as their large subunit counterparts, thus, this is not unusual when working with phages that differ from all the other ones so much. As is usual for cases where no two terminase subunits coding ORFs can be identified in a phage genome, and given the reliable identification of ORF product having TerL feature – the corresponding ORF product was labeled just a “terminase”. We can be sure that this is terminase based on a reliable presence of characteristic conserved domain (near full-length CD search hit to pfam05876 with E-value of 9.85e-17; several confident hits to other phage TerL structures from PDB using HHpred), although it is quite unusually longer than the majority of the TerLs from other phages (e.g. only ~20% of the sequences from the representative dataset are longer than 600 aa, with only several of them being longer than terminase from Mecenats66 (>873 aa). However, even if the TerS would be identifiable – it would be even more problematic to perform MSAs, etc. with TerS sequences due to their way lesser evolutionary conservation than that of TerL (which also tend to form clades very distinct one from another). Thus, although TerS sequence examination was not possible in this case due to our inability to identify it reliably, any considerable insights based on its sequence would not very likely be warranted given the divergence of Mecenats66 protein aa sequences from proteomes of other phages uncovered so far and studied in sufficient detail.

Although we have failed to comply with the suggested improvements of the manuscript, we are grateful that you had taken your time to read through and comment on it. This review has, importantly, shown us another interesting bioinformatic task worth trying to perform. We have tried to explain extensively above, why we do not see the possibility to include such things herein, and we are eagerly awaiting the second round of revisions to maybe get some specific suggestions that you would still insist on including within this manuscript if there are any. As well, we are gladly welcoming comments on the briefly proposed methodological workflow that we believe might help to begin trying to answer these questions in a proper way (how particular aa sequence features impact the packaging strategy in an evolutionary context), which we consider performing and publishing elsewhere as a stand-alone paper in the future in case of finding likeminded researchers with complementary expertise interested in this topic. Thank you!

Reviewer 2 Report

The manuscript of Zrelovs et al. describes a novel bacteriophage of Morganella sp. Although the content is interesting there, need to be done some significant corrections. 

This novel bacteriophage is supposed to have a novel, distinct packaging strategy, which was proven in silico and experimentally with other sequencing methods. In the abstract, this method is named web-lab techniques, but I am not sure if this can be called web-lab technique. Besides, it is unclear what the principle of the method under 2.8 is.

I also think that the English must be corrected throughout the text, and the section Results and Discussion has to be simplified; it would be easier to follow the text if some tables which are now supplementary were transformed into the standard tables, especially those presenting the results of annotation analysis.

Text in Figures has to be rechecked; some copies of the same description are twice in some places. 

Author Response

Reviewer 2: The manuscript of Zrelovs et al. describes a novel bacteriophage of Morganella sp. Although the content is interesting there, need to be done some significant corrections. 

Author reply:

We thank the respected Reviewer 2 for his/her suggestions aimed at improving the manuscript. Please, find our replies below. All the relevant changes to the manuscript text (*.docx version) are now made using the track changes for easier localization.

Reviewer 2: This novel bacteriophage is supposed to have a novel, distinct packaging strategy, which was proven in silico and experimentally with other sequencing methods. In the abstract, this method is named web-lab techniques, but I am not sure if this can be called web-lab technique. Besides, it is unclear what the principle of the method under 2.8 is.

Author reply: 

As correctly pointed out, the moniker "wet-lab techniques/methods" is present in the abstract, introduction, and results and discussion (lines 20, 142, and 735, respectively). We agree that this might be considered a rather subjective interpretation as the meaning of "wet lab" is not standardized across the labs. We have opted to use this as an indication that particular things (restriction analysis using NAGE for the selection of appropriate restrictases (e.g. Figure 5, Supplementary Table S2), and further digestions of the genomic phage DNA before Sanger-based sequencing with custom primers) regarding packaging strategy elucidation of Mecenats66 were done in vitro rather than purely in silico/in the "dry lab". However, although helpful to disambiguate what was performed on the bench and what was performed using the laptop, we see how it might not substantiate the usage of "wet lab" in this case, as the analysis performed is rather simple in terms of methodology.  Thus, as per your suggestion, we have opted to: I) change "verified using wet-lab techniques" to "verified experimentally" in lines 19-20 of the manuscript (abstract); II) change "generalized wet-lab methods" in line 142 of the introduction to "generalized experimental methods"; III) remove "wet lab" altogether from the lines 735-736 of the result and discussion section (initially written as " further wet-lab experimental verification". 

Speaking of method 2.8, we are sorry if this has caused your confusion, we have opted not to elaborate on it up to the finest details, but just provide a reference (reference 27 in the manuscript, lines 330-331 include "putative “pac” fragment; see [27] for detailed elaboration on the rationale) to a chapter in a methodology book that provides generalized protocols for the purpose and presents the rationale behind them. Basically, the 2.8. section of our manuscript is a specialized adaptation and extension of the "3.3 Headful DNA Analysis" section of reference 27. 

Reference 27: Casjens, S.R.; Gilcrease, E.B. Determining DNA Packaging Strategy by Analysis of the Termini of the Chromosomes in Tailed-Bacteriophage Virions. In Bacteriophages: Methods and Protocols, Volume 2: Molecular and Applied Aspects; 2009; pp. 91–111.

Below we try to provide a point-by-point explanation for the purpose of the review. If the respected reviewer 2 feels that the section would benefit from such/similar explanation in addition to giving the reference, we will not object to making it more detailed during the second round of revisions, although we are not sure it is necessary given the reference and listing the details specific to our analysis with Mecenats66 in the section.

The section begins with a sentence introducing custom primers that were designed based on the in silico PhageTerm re-organized genome taking the possibility of headful packaging with a defined packaging series initiation site into account. However, this was not supported by the common dataset terminase aa sequence phylogeny. As our NGS library was prepared using sonication as the means of fragmentation (lines 218-220), the randomly generated genomic fragment DNA read pile-up pattern characteristics consistent with such packaging (as determined by PhageTerm) were very unlikely to happen by chance alone, highlighted the necessity to investigate this observation further.

In the case of headful packaging phages (assuming the genome representation used to design the primers to hybridize somewhere in the proximity of the exact termini is organized correctly) it is to be expected to get Sanger reads that span further than the expected genome termini with the respective chromatogram peaks being of high quality (if the primer is designed appropriately) for the undigested headful packaging employing phage genomic DNA. This can be seen in tracks 2 and 3 of Figure 6. However, note, that the -1 base of the genome representation had a double peak, having a minor "A" peak in addition to the called "G", while all the other peaks had unambiguous base calls. Although it is possible to wave this away as an SNP, it was present only in track 2, but not 3. The location of this small "A", however, is consistent with the NGS read pile-up analysis pattern determined (PhageTerm) genome terminus/concatemer packaging series initiation site. Thus, the evidence points that this might be a non-template single base overhang added by the polymerase.

However, to be sure that we are working with the termini region, a restriction analysis was ought to be performed. For that, we wanted to select restrictases that would display the DNA pac fragment at an uncrowded gel position, as per reference 27. This was easily done based on the de novo assembled and PhageTerm reorganized genome by performing in silico digestions of the molecule using all the restrictases available in the lab and selecting the ones that are fit for our purpose. Thus the second paragraph of section 2.8. includes mention of the digestion with particular restrictases "which were determined by in silico digestion to result in the digestion profiles that would present an identifiable fragment resulting from a cut near the physical beginning of the genome (putative “pac” fragment;". To avoid a rather lengthy explanation of the rationale, reference 27 is given in line 330. However, we feel that we should maybe more specifically point to sections 1.3. and 3.3. within that reference, which is now done in the second version of the manuscript.

The third paragraph indicates that after identification of the putative "pac" fragment-containing band - it was cut out from the gel and sequenced using the appropriate primer. We, however, see that despite pointing the reader to reference 27, this chapter misses mention of the native agarose gel electrophoresis for the restriction fragments in this section, although it should have been present initially. This was a mistake, and the second paragraph of section 2.8 now includes additional sentences that clarify the performance of NAGEs. The following sentences were added:

"Native agarose gel (1%) electrophoresis was performed loading digestion reactions of Me-cenats66 genomic DNA with each of the restrictases used into individual wells and using GeneRuler 1 kb Plus DNA Ladder (Thermo Fisher Scientific) in a marker well. After visu-alizing the generated restriction profiles in the gel, the SmiI-generated profile was selected as the most convenient one to work with further. Thus, a larger amount of phage genomic DNA (~5 µg) was additionally digested with SmiI. Subsequently, NAGE was performed to visualize the restriction fragments and extract the fragment of interest from the gel."

The rationale behind this lies in the fact that the faint band that was seen is very likely to contain the phage Mecenats66 genomic DNA concatemer packaging series initiation site. This, indeed, shows that the fragments of somewhat varying length were cut near the genome termini, however, the fragments of precise length dominated the appropriate restriction fragment pool near the genome terminus, implying a preferred packaging series initiation site in contrast to it being initiated completely randomly. When this fragment was cut out from the gel and sequenced using the appropriate primer and chromatograms mapped unto the junction of Mecenats66 genome representation, the characteristic features of headful packaging with a preferred packaging series initiation site were evident. This is seen in figure 6, track 4. There, we can see that this "pac fragment" indeed had a defined terminal base, after which the polymerase used was adding the non-template base. Additionally, the weak signal for peaks spanning this terminal base and corresponding to the second terminus can also be seen, which are a result of imprecision during manual excision of the corresponding fragment from the gel, excising some volume of the gel slightly below or above the fragment of interest (that contains fragments of other lengths corresponding to the ones cut from the genomes that come from virions that packed their DNA within the procapsid in a headful manner later than in the first concatemer packaging event that is defined precisely). The rationale for this experiment is visually shown very well in Figure 7.1. of reference 27, depicting a schematic representation of "P22 headful packaging".

We hope that we were able to explain the principle behind this method to the respected reviewer. We also hope that some clarifying sentences added to section 2.8. of the initial version of the manuscript now make it clearer. 

The same information that shows how appropriate pieces of evidence gradually inform the further actions then to yet be performed are described in "3.3. Exact genome termini and genome packaging strategy elucidation for Mecenats66" of the results and discussion that is written in a similar way to our explanation here, but presenting the results of this workflow in a more detailed, yet formal and concise manner as is expected for a scientific manuscript. This section, we hope, aids in understanding the principle of the methodology used by discussing the obtained results.

Reviewer 2: I also think that the English must be corrected throughout the text, and the section Results and Discussion has to be simplified; it would be easier to follow the text if some tables which are now supplementary were transformed into the standard tables, especially those presenting the results of annotation analysis.

Author reply: 

We thank you for pointing to the language improvements necessary, all the text has now been re-read, and, indeed, some of the minor spelling/grammar mistakes were spotted and corrected. Although neither of the authors is a native English speaker, we tried our best at re-checking the text and improving the language. Should some mistakes still remain, we hope that these can be resolved in the typesetting stage in case of the acceptance of the manuscript for publication.

As for the Results and Discussion section, we do not understand what would require further simplification. All the information included in the section reinforces the conclusions. Removing some of it might do more harm than good for the understanding of the work we've carried out by the readers. Currently, the section has four subsections. 

The first one is rather concise and introduces the host, which currently looks like it might represent a novel species of Morganella, but this requires further experimental work to be done and published elsewhere.

The second subsection is also a rather short one and introduces Mecenats66 virion morphology and its genome. The suggestion to make the supplementary table with detailed annotation of the Mecenats66 genome a regular table within the main text is interesting. Although we have seen that done in some phage characterization papers previously, personally, we had a previous experience when putting such a massive (both vertically and horizontally) table within the main text that such move was actually spoken against by the referees so as not to take up so much space in the body of the manuscript if it can be condensed in a few text paragraphs and added as supplementary material for more interested readers to which they are referred throughout the text. Although we are used to including the detailed genome annotation tables within our phage papers in the supplementary materials, we stay neutral in regards to the implementation of this suggestion and will not object to its inclusion in the main text if this will be allowed by the journal team. Currently, we did not implement this change, as this would mean reducing the text size of the table contents to the point that would make it unreadable without zooming in on it, but if the reasoning we've told about previously does not seem appropriate, we will try to do so in the second round of revisions in case you insist and journal team supports it.

We are not sure this suggestion, however, is applicable for Supplementary Table 2, which provides an "expected vs observed" restriction patterns. While it was added to supplement evidence for our results and discussion, Figure 5 showing what was observed is more important, and adding Supplementary Table 2 to the main text would be redundant.

As for Supplementary table 3, we believe it is completely irrelevant to add to the main text, as it only lists the names and accessions of the amino acid sequences used for the compilation of the "representative TerL diversity" dataset, MSA, and phylogenetic tree generation, and is of interest only to those readers wanting to repeat the analysis/get the same dataset for their own endeavors, rather than general readership. 

Subsection three is a lengthier one, actually, including the results we find to be one of the most interesting ones within the manuscript. The gradual evidence build-up presented herein retains the original experimental "flow" and allows for an understanding of the methodology employed for packaging strategy analyses and will likely be of value for readers that want to use these approaches on their own study objects. The closing paragraph of the subsection also shows some of the in silico prediction incongruencies between the mechanistic PhageTerm inference details of the same phage read analysis from two independent library preps, which might be of interest to people using this software.

Subsection four shows a broader context of TerL/terminase aa sequence phylogeny based approach for packaging strategy prediction. It places the Mecenats66 terminase aa sequence (as well as the common dataset with phages that had their packaging strategies experimentally verified) in a representative TerL/terminase diversity - showing that it indeed is distinct even in such a context. And discusses the implications of the fact that many of the phages sequenced do not have their genome termini/packaging strategy identified. 

As mentioned in the cover letter, we also ask to note that some of the (supplementary) figures now present in the main text were there to make it easier for the respected editor and reviewers to follow the points being made in the main text, thus, they are not ought to be present in the final version, and their absence will simplify the contents of result and discussion section. If the respected reviewer 2 would point to specific lines that are in his/her opinion in need of simplification, we will try to rewrite them in a more simple manner during the next round of revisions, but we could not identify such parts (although, this too is subjective, and the manuscript, obviously, was written by us in a manner we find simple enough).

Reviewer 2: Text in Figures has to be rechecked; some copies of the same description are twice in some places. 

Author reply:

We believe this is applicable only to figures 3 and 7. Figure 3 shows pairwise genome organization and encoded protein comparison of Morganella phage Mecenats66 (middle) to Pseudomonas phage Dolphis (upper), and to Bordetella phage vB_BbrM_PHB04 (lower). The caption includes a line "Slanted labels above the arrows indicate the predicted function for the given ORF putative product in case it had a function assigned". Three genomes are compared, and we have decided to retain respective ORF product labels that were given by the original annotated genome submission authors prior to putting it into the public biological sequence repository. Thus, in the case of figure 3, it is expected that some of the labels (e.g. DNA primase/helicase, portal protein, MCP are seen in double or triple) as each one is applicable to a particular phage genome ORF. This may seem redundant if the function of the homologous ORF in all three of the genomes is predicted to be the same/similar, but that way it allows us to visualize the differences in the annotation of the homologs and the presence/absence of particular proteins in one or more of the genomes compared. We, however, have noted that, sadly, the resolution of Figure 3 is lackluster as something has gone wrong during the conversion/uploading process (in both pdf and docx). This was, of course, not the intended quality, as the ribbon labels that indicate the pairwise similarity between the connected ORF products are unreadable even when zoomed in. We have tried to update Figure 3 in a better resolution with the hope that it will get converted/uploaded properly. We are sorry for this technical issue, and we hope that this is now resolved in the second version - if not, we will try to resolve this with assistance from a journal team.

However, most likely this is the reference to Figure 7 showing the Maximum-likelihood tree of representative terminase/terminase large subunit (TerL) amino acid sequence diversity. In this figure indeed, some of the tip labels are seen up to five times. This, however, was intentional and is indicated in the following line of the figure 7 caption: "Tip labels are colored based on the distinct packaging strategies phages within the clades presumably employ (sequences from the extended “core” dataset have their respective packaging strategy at the corresponding tips [34]; Figure 4)". However, we see how this might be puzzling and, thus, opt to modify the caption. The goal was to put these labels indicating the packaging strategy for tips from the phages that had their packaging strategy/genome termini type verified experimentally. At first glance, it may seem redundant when we designate clades using everything branching out of the MRCA of sequences that includes all of these verified phages in a given clade (from Fig 4) and one label might be given for a clade altogether by extension. However, having labels indicating the packaging strategies for such "experimentally verified" leaves allows the reader to better "read" the tree and see the distances between the verified representatives in the tree, and to disambiguate them from TerL/terminases of phages that presumably employ the same packaging strategy based on this clustering (also shown by the black dotted lines from the leaves to the outer radius of the tree; accordingly colored dotted lines for experimentally verified phages from the corresponding leaf to label). For clarity, the aforementioned part of the caption was rewritten to include "Tip labels are provided for phages that had their packaging strategy/genome termini type verified experimentally and are colored based on the distinct packaging strategies phages within the presumed same packaging strategy clades employ (sequences from the extended “core” dataset have their respective packaging strategies indicated at the corresponding tips [34]; Figure 4)" for better clarity.  Although we have tried different visualization approaches, visualizing such a tree in a meaningful way while retaining some of the associated sequence annotations we wanted to retain proved itself not to be a trivial task (e.g. color coding alone is not enough when so many categorical variables are present - adding an outer annotation radius with shapes will not be comprehensible at a glance; also, there is a hindrance that some of the tips needed to be further disambiguated from the others in a tree with so many leaves, where tip shapes would not be observable without zooming in much). This was the best visualization option we've arrived at after numerous tries, although it contains some redundancy, we did not find a better way to make the tree comprehensible at a glance. However, if you happen to have some visualization suggestions for this particular case that were not tried by us - we will be glad to try them out.

We are thankful to the reviewer for reading through and commenting on the manuscript. Although we have tried our best to implement the suggestions of reviewer 2 or explain ourselves if the suggestion was not a straightforward one to implement, should anything still remain inappropriately addressed or unexplained, we are eagerly looking forward to the second round of revisions, to take note of the more specific suggestions that might further improve the manuscript in case of necessity.  Thank you!

Reviewer 3 Report

Impressive study showing how important is to have researchers team following any potential opportunity for a new discovery.
I only comment about the `Introduction` that could be re-organized and reduced (even it is very interesting - but is somehow like a review, itself; some information being well-known)

Author Response

Reviewer 3:

Impressive study showing how important is to have researchers team following any potential opportunity for a new discovery.

I only comment about the `Introduction` that could be re-organized and reduced (even it is very interesting - but is somehow like a review, itself; some information being well-known).

Author reply: 

Thank you for the kind words! It indeed was observation-driven research further fueled by our curiosity.

Regarding the introduction, this "review-like" writing style for it was used intentionally. MDPI's Microorganisms "instructions for the authors" state that the introduction, among other points, should be "comprehensible to scientists working outside the topic of the paper" while placing the study in a broad context, and reviewing the current state of the research field including citing the key publications. We undeniably agree that this "comprehensible to scientists working outside the topic of the paper" part can be understood differently, but below we will try to give our rationale for writing the introduction as seen in the initial version of the manuscript.

The introduction includes eight paragraphs and one table, which makes it a rather lengthy one. The main struggle for us was to retain the easy flow of this section while trying to connect two main topics of the paper (e.g. introducing background not only for the reader to get a glimpse into the research of rare bacterial host phages, among which Morganella sp. phages happen to be but also to provide information about packaging strategies phages employ to allow the reader to understand the major part of the paper). Each of the paragraphs, however, has its own purpose, as we see it (described below). 

The first paragraph introduces the point that "the known cultured phage diversity is still largely restricted to a relatively limited number of distinct hosts" and points to the expected growth in the importance of phages in healthcare. The first paragraph also informs the reader on the particular selected bacterial genera implicated in health, which, in our opinion, might benefit from research on their phage diversity, that is currently rather limited or lacking for such hosts so far. Thus, we hope that some of the readers working with phages might get inspired to give some of these selected hosts a try for their future phage hunting.

The second paragraph gives a closer look on the genus of host bacteria (Morganella) to which host of phage Mecenats66 belongs. This paragraph also shows that Morganella species and their phages are of (although not yet immediate) potential interest for healthcare. The fact that the genus Morganella so far comprises only two bacterial species presented in this paragraph allows the reader to further appreciate that Morganella sp. B-1 (host of Mecenats66) derived from bees differs from the so far recognized Morganella species, and might represent a novel bacterial species within the genus itself (although this was not elaborated much within the paper as far more additional experimental work is necessary to show and propose this).

The third paragraph introduces the so far uncovered Morganella phage diversity, showing that it is, indeed, rather limited at the time. Some of the information of interest is also condensed in a table that could prove useful for readers interested in working with the phages of Morganella. This concludes the "microbiological" part of the introduction (first three paragraphs) giving background on/highlighting the importance of obtaining phages for the less-commonplace hosts, host genera of Mecenats66, and diversity of the phages infecting the same host genera. Basically, these paragraphs tell the reader why the phage described in the paper is "not yet another novel phage", and why it seemed interesting for us to study.

Paragraphs four to six (including) of the introduction, indeed, are composed partly of somewhat "textbook knowledge". However, they were written with a goal and in a way that, we believe, allows to concisely introduce a rather specific topic of dsDNA bacteriophage genome packaging to most of the readership of Microorganisms journal that are not specifically focusing their research on phages (brief mention of the protein players in paragraph four, genome termini type variety in paragraph five, and closer look at the headful packaging strategy and its implications in paragraph six). We hope that these paragraphs and the references cited within them will be able to either refresh or give enough of the background on this subject for readers out of this specific area of investigation to understand the contents of the paper. Likewise, references and examples cited in this part of the introduction serve as a good starting point for further reading regarding phage genome packaging and termini types.

Paragraph seven shows that despite the availability of different methods to scrutinize novel phage genome termini types and, thus, packaging strategy, this was so far done for a relatively small number of phages that have had their complete genomes elucidated. Thus, importantly, this paragraph introduces both, the recurrent problem in phage genomics and possible solutions, of which many readers might be unaware.

Paragraph eight concludes the introduction and connects the previous paragraphs composed of background relevant to the understanding contents of the manuscript directly to the work that was carried out by us. Importantly, the closing paragraph of the introduction highlights that TerL/terminase amino acid sequence phylogenies might not give an unambiguous answer about what packaging strategy/termini a novel phage employs.

That being said, an introduction to the microbiological background might not flow well enough into the more specific topic of tailed phage genome packaging, both of which are integral for understanding the contents of the paper (composed of both introducing a novel phage of an interesting host, and scrutinizing its genome termini/packaging). Despite understanding this, we were unable to come up with a way that would allow us to seamlessly glue these parts together within an introduction. So far, the only workaround we imagine for "re-organization" of an introduction might be trying to better separate both topic backgrounds provided within it by adding distinct headers to paragraphs 1-3 (including the table), and paragraphs 4-8. However, we are not really sure that would really benefit the manuscript, as currently Table 1 serves as their separator of a kind.

Above we have tried to explain our rationale for choosing to write the introduction the way it was seen in the initial submission. We believe that it is in line with the "Research Manuscript Sections - Introduction" guidelines for the authors by the journal and that reducing it (which would mean omitting some of the references to the important work being done by other researchers elsewhere) might harm comprehensibility of the manuscript in its entirety to the general readership of Microorganisms. That said, if the respected Reviewer 3 disagrees with our rationale, we would be happy for suggestions on what specifically might, in your opinion, be rewritten in a more concise way or removed altogether without harming the comprehensibility of the manuscript. Although we would like to see the introduction the way it is, we are ready to try improving this section as per your more specific suggestions in the next round of revisions should you find our reasoning inappropriate. Thank you!

Round 2

Reviewer 1 Report

I would like to thank the authors for their thoughtful response, and I concur that the manuscript, as written, is suitable for publication.

Reviewer 2 Report

The manuscript has been corrected accordingly.